# Clinical Results of a Modified Doty’s Technique for Supravalvular Aortic Stenosis

**DOI:** 10.3390/jcm12051731

**Published:** 2023-02-21

**Authors:** Lizhi Lv, Xinyue Lang, Simeng Zhang, Cheng Wang, Yuanhao Jin, Qiang Wang

**Affiliations:** 1Center for Pediatric Cardiac Surgery, Fuwai Hospital, National Center for Cardiovascular Diseases, Chinese Academy of Medical Sciences and Peking Union Medical College, Beijing 100037, China; 2Department of Cardiac Surgery, Yunnan Fuwai Cardiovascular Hospital, Kunming 650102, China; 3Medical Research & Biometrics Center, National Center for Cardiovascular Diseases, Fu Wai Hospital, Chinese Academy of Medical Sciences and Peking Union Medical College, Beijing 102308, China; 4Department of Cardiac Surgery, Peking University People’s Hospital, Beijing 100044, China

**Keywords:** supravalvular aortic stenosis, surgical repair, modified Doty’s technique, Doty’s technique, surgery-related complications

## Abstract

This study aimed to assess the early and mid-term results of the modified Doty’s technique compared with the traditional Doty’s technique in patients with congenital supravalvular aortic stenosis (SVAS). We retrospectively included 73 consecutive SVAS patients in Beijing and Yunnan Fuwai Hospitals between 2014 and 2021. Patients were divided into the modified technique (*n* = 9) and the traditional technique group (*n* = 64). The modified technique involves altering the right head of the symmetrical inverted pantaloon-shaped patch into an asymmetrical triangular form to prevent compression of the right coronary artery ostium. The primary safety outcome was in-hospital surgery-related complications and the primary effectiveness outcome was re-operation at follow-up. The Mann–Whitney U test and Fisher’s exact test were used to test the group difference. The median age at operation was 50 months (IQR 27.0–96.0). Twenty-two (30.1%) of the patients were female. The median follow-up was 23.5 months (IQR 3.0–46.0). No in-hospital surgery-related complications and follow-up re-operation occurred in the modified technique group, but the traditional technique group had 14 (21.8%) surgery-related complications and 5 (7.9%) re-operation. Patients with the modified technique had a well-developed aortic root and no aortic regurgitation occurred. A modified technique could be considered for patients with poor aortic root development to reduce postoperative surgery-related complications.

## 1. Introduction

Supravalvular aortic stenosis (SVAS) is a congenital cardiac malformation that accounts for 8–14% of all congenital aortic stenosis [1,2]. It is defined as discrete stenosis (type I) or diffuse stenosis (type II) of the aorta located above the aortic valve [3]. The natural disease process of SVAS is progressive and will lead to obstruction of the left ventricular outflow tract, resulting in distal tissue ischemia and left ventricular hypertrophy. Early treatment is extremely important for SVAS [4]. 

Doty’s technique was generally used to treat SVAS by relieving obstruction at the aortic root and sinotubular junction (STJ) [5,6]. However, myocardial ischemia with inadequate perfusion occurred in some cases post-procedure which increased the risk of perioperative malignant arrhythmias, perioperative ECMO assistance, and secondary open-heart surgery [1,2,7,8,9,10,11,12,13]. We surmised it is caused by the inappropriate shape of the patch. Doty’s technique included a patch sutured into the right coronary sinus, and the rounded head of the patch might lead to displacement and compression of the right coronary artery ostium, resulting in a fold of the right coronary artery ostium or proximal end. 

Our center had changed the right round head of the pantaloon-shaped patch to a triangle shape in two SVAS cases treated by traditional Doty’s technique with frequent ventricular arrhythmias 1 day after surgery. Neither patient developed malignant arrhythmias or severe low cardiac output after the modification of the patch [8]. In this study, we aimed to further explore the effectiveness and safety of the modified technique (right triangle head) compared with the traditional technique (right round head) for the early and mid-term prognosis.

## 2. Patients and Methods

### 2.1. Patient Population

The clinical data of consecutive patients with the modified technique and the traditional technique were retrospectively analyzed in the electronic medical record system of Beijing Fuwai Hospital and Yunnan Fuwai Hospital. Patients who underwent modified Doty’s technique and traditional Doty’s technique between May 2014 and December 2021 were included. The SVAS were diagnosed by a trans-thoracic echocardiogram (TTE). Patients treated with other repairs (McGoon’s technique, Brom’s technique, the sliding aortoplasty, and Ross surgery) or who underwent second surgery were excluded (Figure 1). The study protocol was conducted by Fuwai Hospital Human Subjects Committee guidelines and was approved by the Institutional Review Board (no.2021-1578). The application for waiver of informed consent was also approved. This study was registered at www.chictr.org.cn (accessed on 16 May 2021) (ChiCTR2100046494).

### 2.2. Operative Technique

Cardiopulmonary bypass was used in all cases after median sternotomy and cannulation of the ascending aorta. Cardiac arrest was achieved with Custodiol^®^ HTK cardioplegia solution. The cases were divided into traditional techniques (Doty’s technique) and modified techniques. For the traditional technique, a longitudinal incision through the stenosis was extended to the right coronary sinus and noncoronary sinus, respectively, a traditional pantaloon-shaped patch was inserted, and the narrow part was widened. For the modified technique, the aortic root incision was the same as in the traditional technique, and we changed the right head of the symmetrical inverted pantaloon-shaped patch to an asymmetrical triangular shape to avoid the compression of the right coronary artery ostium (Figure 2). 

### 2.3. Variables and Outcomes

The variable of baseline information included operation age, gender, body surface area (BSA), Williams syndrome (WS), type of SVAS, echocardiographic variables (supravalvar aortic gradients, stenosis diameter, z-score of aortic valves, STJ, and ascending aorta) and concomitant malformation. The echocardiographic data of all available echocardiography were analyzed by two cardiologists (Q.W. and C.W.). Z-score was calculated through Boston Children’s Hospital echocardiography calculation tool (https://zscore.chboston.org/) (accessed on 22 November 2021). The surgery-related variables included surgical technique, patch material, cardiopulmonary bypass time (CPB), and cross-clamping (CCP) time. The patch material included pericardial patch (fresh autologous pericardium), artificial material (Dacron^TM^, Rastatt, Germany; artificial vascular patch^TM^, LLC, Arizona, USA, W. L. Gore & Associate (Newark, DE, USA; bovine pericardium^TM^, Zhejiang, China, JHZB Biotech group; and homogeneous allogeneic vascular patch, home-made) and modified patch (glutaraldehyde-treated autologous pericardium and fresh autologous pericardium outer lining with artificial material).

The primary safety outcome was defined as surgery-related complications due to myocardial ischemia during the operation or within 24 h post-operation [14] (including repeated aortic clamping, cardiac defibrillation, delayed chest closure, extracorporeal membrane oxygenation (ECMO) needed, acute myocardial ischemia, and malignant arrhythmias). The primary effectiveness outcome was the re-operation at follow-up. Each patient’s pre-operative and post-operative cardiac anatomy was evaluated with echocardiography. Restenosis at follow-up was defined as mean supravalvar aortic gradients over 40 mmHg [12]. Early reoperation was defined as any operation in the hospital or within 30 days.

### 2.4. Statistical Analysis

Categorical variables were described by the frequency (percentage). Continuous variables were described by mean ± SD and median (inter-quartile range (IQR)). The group difference was tested by the Mann-Whitney U test for the continuous variables. Fisher’s exact test was used to compare the categorical data. A two-sided *p*-value < 0.05 was considered significant. All analyses were conducted using R software (version 4.0.3).

## 3. Results

### 3.1. Baseline Information

A total of 73 patients were included in this study. The median age at operation was 50 months (IQR: 27.0–96.0). A total of 30.1% (22) of the patient was female. Nine patients underwent the modified technique, and sixty-four patients underwent the traditional technique. For the modified technique, three (33.3%) were women, and one (11.3%) patient was WS. For the traditional technique, 19 (29.7%) were women, and 6 (9.4%) patients were WS. Patients who used modified techniques (median: 84.0, IQR: 27.0–168.0) were older than patients who used traditional techniques (median: 50.0, IQR: 31.5–95.0). Modified technique patients also had a lower aortic valve z-score (median: −1.3, IQR: −2.3, −0.8) and STJ z-score (median: −0.4, IQR: −1.1, 0.30) compared with the patients who used the traditional technique (aortic valve z-score, median: −0.5, IQR: −1.4, 0.2; STJ z-score, median: 0.9, IQR: −0.2, 1.9). The ejection fraction (EF), supravalvar aortic gradients, and ascending aorta z-score were similar in the two groups. Additionally, concomitant malformation had no difference in the two groups (Table 1).

### 3.2. Intraoperative and Postoperative Outcomes

In terms of patch material use, pericardial patch (5 (55.6%)) and modified patch (4 (44.4%)) were used in the modified technique group, whereas in the traditional technique group, the autologous pericardium was used the most (37 (57.8%)), followed by modified patch (15 (23.4%)), while artificial materials were used the least (11 (17.2%)). No significant difference was found between CPB and CCP. 

For surgery-related complications, no case occurred in the modified technique group, and 14 cases occurred in the traditional technique group, 11 of which had ventricular fibrillation during cardiac resuscitation, 1 case had repeated aortic block, 1 case had repeated aortic block combined with malignant arrhythmias, and 1 case had repeated aortic block combined with postoperative ECMO assistance and delayed chest closure. For postoperative EF, the modified technique was similar to the traditional technique (median: 65.0, IQR: 63.0–70.0 vs. median: 65.8, IQR: 63.7–70.0). For postoperative supravalvular pressure gradients, the modified technique group was lower than the traditional technique group (median: 16.0, IQR: 8.4–18.7 vs. median: 19.7, IQR: 10.2–31.4). No significant difference was found in the *z*-value of aortic valve, STJ, and ascending aorta in the two groups (Table 2).

### 3.3. Follow-Up Outcomes

The echocardiographic follow-up was conducted in 98.6% (72/73) of patients. The median follow-up duration was 23.5 months (IQR: 3.0–46.0). The mid-term follow-up results in the modified group versus the traditional group were favorable (Figure 3). No death occurred, and only one patient in the modified technique group had an elevated supravalvular pressure gradient during follow-up, and this case had a preoperative diagnosis of diffuse SVAS with both WS and aortic valve stenosis. For the primary effectiveness outcome, the modified technique group had no re-operation, while five (7.9%) re-operations occurred in the traditional technique group. No significant difference was found in terms of EF, supravalvular pressure gradient, and restenosis between the two groups. In addition, there was no difference in the growth of the aortic root during the follow-up period (Table 3).

## 4. Discussion

This retrospective clinical study evaluated the early and mid-term prognosis after the modified technique for congenital SVAS from 2014 to 2021 and compared it with the traditional technique. Among nine patients who used the modified technique, no in-hospital surgery-related complications and follow-up re-operation occurred, while the patients who used the traditional technique had fourteen (21.8%) surgery-related complications and five (7.9%) re-operations. Meanwhile, the growth of the aortic root after the modified technique was non-inferior to the traditional technique. 

McGoon et al. [15] first came up with the single-patch repair for congenital SVAS in 1961. In 1977, Doty et al. [5] described the pantaloon-shaped patch that extended into two valve sinuses. After that, Brom [6] reported a technique with three separated and symmetrical patches into three aortic valve sinuses. This change aimed to reconstruct a symmetrical ascending aorta. Compared with McGoon’s technique, Doty’s technique and Brom’s technique could repair the aortic stenosis more symmetrically and completely [16,17,18]. However, these two types of multi-sinus autoplasty influenced the form of the coronary sinus. The location and depth of the incision, the material of the patch, and especially the size could also affect the coronary blood flow [1,7,8,19,20,21]. 

When the pantaloon-shaped patch is sutured in the noncoronary sinus and right coronary sinus using the traditional Doty’s technique, as well as when Brom’s technique is used for three sinus patching, it leads to coronary artery distortion obstructing coronary flow and inadequate perfusion, increasing the risk of perioperative myocardial ischemia. Delius et al. [1] reported a case of inadequate perfusion of the left anterior descending branch after Brom’s technique, where intraoperative displacement of the left coronary opening was found to distort the proximal left anterior descending branch, which affected the left anterior descending branch perfusion, which was successfully weaning from cardiopulmonary bypass after intraoperative repatching. Ramakrishnan et al. [7] also found a distortion of the proximal right coronary artery after Brom’s technique of SVAS, and they restored coronary perfusion by patches with some strategic horizontal mattress sutures. Recently, Luo et al. [20] found that right coronary artery distortion and kinking were always caused by the bulging of the adjacent redundant longitudinal dimension of the pantaloon-shaped patch, which shifted the proximal right coronary artery anteriorly, inferiorly, and laterally. Thus, they proposed an H-patch technique for SVAS, using two separate oblong patches for the reconstruction of the noncoronary and right coronary sinuses, thereby preventing displacement of the right coronary ostium.

In our center, the chief surgeon in our team began to use this modified technique to repair SVAS starting in 2014. This technique has now become a standard procedure for patients with discrete SVAS and severe obstruction. However, for diffuse SVAS, the additional aortic arch repair is necessary in conjunction with this technique. A total of nine patients have received the modified technique, with no operative death, operation-related complication, and follow-up reoperation. The early and mid-term prognosis of these nine patients was able to show that the modified technique was an anatomically and technically effective surgical approach to treating SVAS. What is more, this technique is a modification of Doty’s technique, the operator does not need to make a new incision and additional patches only need to adjust the original patch, which can effectively reduce the occurrence of surgery-related complications, while ensuring the growth of the aortic root.

Our study introduced the modified technique in detail. Although no statistical significance was seen in the primary outcomes, the differences in values between the two groups may suggest clinically relevant differences. When compared with the traditional technique, we found lower surgery-related complications and follow-up re-operation, and the follow-up TTE outcome had no significant difference, suggesting that the modified technique was non-inferior to the traditional technique in terms of safety and efficacy. However, our study still has some limitations. This was a retrospective study. Only nine patients underwent the modified technique, and a large sample size is needed to validate the clinical effectiveness of the modified technique in the future.

## 5. Conclusions

The surgery-related complications and re-operation rates of the modified Doty’s technique were not higher than that of the traditional technique, and the follow-up trans-thoracic echocardiogram outcome had no significant difference compared with the traditional technique. A modified technique could be considered for patients with poor aortic root development to reduce postoperative surgery-related complications.

## Figures and Tables

**Figure 1 jcm-12-01731-f001:**
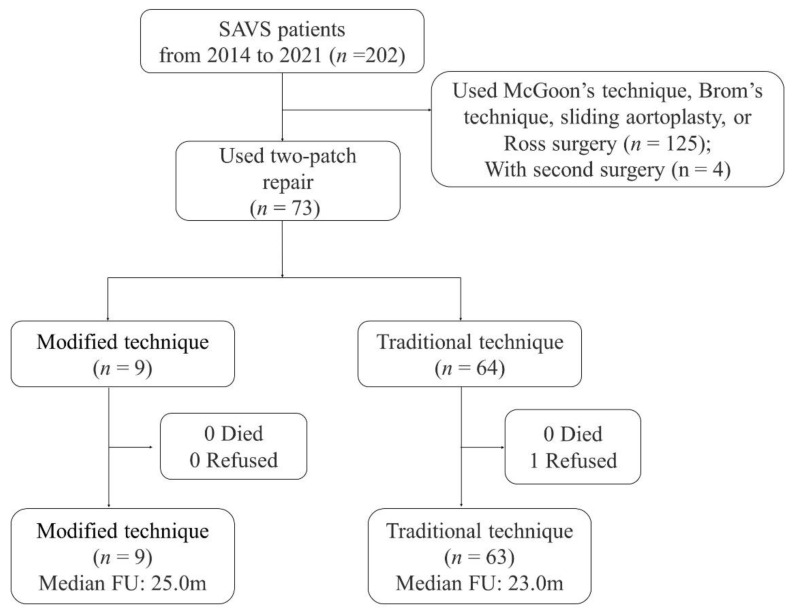
**Flow chart of patient selection and follow-up.** Abbreviation: FU, follow-up; SVAS, supravalvular aortic stenosis.

**Figure 2 jcm-12-01731-f002:**
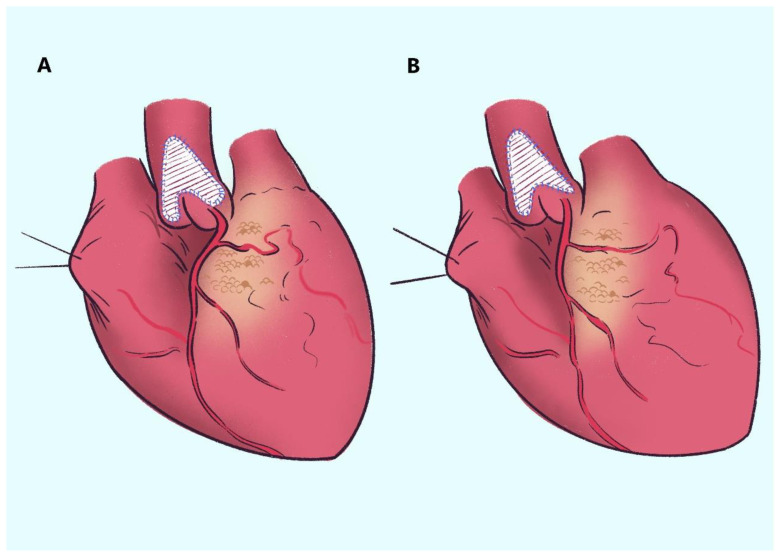
**Adjustment of right coronary sinus patch.** (**A**). Traditional technique. The right coronary sinus distorted after the traditional pantaloon-shaped patch; (**B**). Modified technique. A triangular patch is added to the right coronary sinus, with no displacement or deformation of the right coronary ostium.

**Figure 3 jcm-12-01731-f003:**
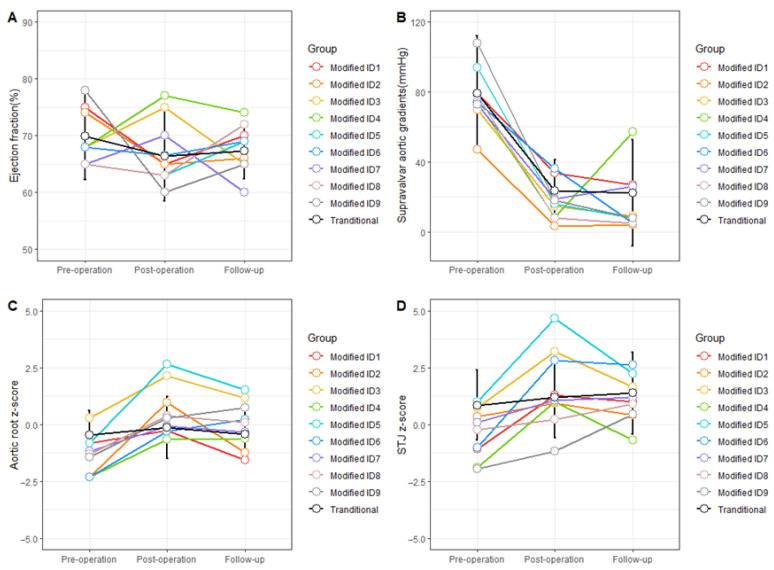
**The information of the modified and traditional technique pre-, post-operation, and at follow-up.** (**A**) Ejection fraction; (**B**) Supravalvar aortic gradients; (**C**) Aortic root z-score; (**D**) STJ z-score. The patient ID1-ID9 with the modified technique are shown in red, orange, yellow, green, cyan–blue, blue, purple, pink, and grey separately. The patient information of the traditional technique is shown in black, and the points mean the mean value, and the bar means mean ± sd. Abbreviation: STJ, sinotubular junction.

**Table 1 jcm-12-01731-t001:** Baseline characteristics of the patients with the modified and traditional technique.

Variables	Modified Technique (*n* = 9)	Traditional Technique (*n* = 64)	*p* Value
Age (months)	121.7 ± 121.8 84.0 (27.0, 168.0)	86.3 ± 105.1 50.0 (31.5, 95.0)	0.6028
Women	3 (33.3)	19 (29.7)	0.8234
BSA (m^2^)	1.1 ± 0.5 1.1 (0.7, 1.7)	1.0 ± 0.4 0.9 (0.7, 1.1)	0.6088
EF (%)	69.6 ± 4.9 68.0 (65.0, 74.0)	69.9 ± 7.7 70.0 (65.0, 75.8)	0.7550
Supravalvar aortic gradients (mmHg)	77.4 ± 16.7 74.0 (73.0, 79.0)	79.3 ± 32.6 75.3 (59.4, 98.0)	0.9933
Diameter of the stenosis (mm)	8.7 ± 2.2 8.0 (7.0, 10.0)	7.9 ± 2.7 7.2 (6.0, 9.0)	0.2874
Aortic valve z-score	−1.4 ± 0.9 −1.3 (−2.3, −0.8)	−0.5 ± 1.1 −0.5 (−1.4, 0.2)	0.0230
STJ z-score	−0.4 ± 1.1 −0.2 (−1.1, 0.3)	0.8 ± 1.5 0.9 (−0.2, 1.9)	0.0176
Ascending aorta z-score	−1.0 ± 2.1 −1.3 (−2.2, −0.1)	−1.4 ± 2.1 −1.9 (−3.0, −0.4)	0.4758
Arrhythmia	1 (11.1)	4 (6.3)	0.5888
LVH	7 (77.8)	39 (60.9)	0.3272
Form of SVAS (discrete)	9 (100)	60 (93.8)	0.4405
WS	1 (11.1)	6 (9.4)	0.8685
**Concomitant malformation**			
PDA	0 (0.0)	2 (3.1)	0.5908
ASD	1 (11.1)	1 (1.6)	0.1004
VSD	0 (0.0)	3 (4.7)	0.5071
ALCAPA	0 (0.0)	1 (1.6)	0.7057
Ascending aortic stenosis	0 (0.0)	4 (6.3)	0.4405
HOCM	1 (11.1)	0 (0.0)	0.0073
PVS	0 (0.0)	1 (1.6)	0.7057
Bicuspid aortic valve	0 (0.0)	1 (1.6)	0.7057
MVR	0 (0.0)	6 (9.4)	0.3377
AVS	0 (0.0)	4 (6.3)	0.4405
AVR	2 (22.2)	14 (21.9)	0.9812
Supra-aortic septum	1 (11.1)	8 (12.5)	0.9055

Abbreviation: ALCAPA, anomalous left coronary artery from the pulmonary artery; ASD, atrial septal defect; AVR, aortic valve regurgitation; AVS, aortic valve stenosis; BSA, body surface area; EF, ejection fraction; HOCM, hypertrophic obstructive cardiomyopathy; LVOT left ventricular outflow tract; MVR, mitral valve regurgitation; PDA, patent ductus arteriosus; PVS, pulmonary valve stenosis; STJ, sinotubular junction; SVAS, supra-aortic stenosis; TOF, Tetralogy of Fallot; VSD, ventricular septal defect; WS, Williams syndrome.

**Table 2 jcm-12-01731-t002:** Intra- or post-operative information of the patients with modified and traditional techniques.

Variables	Modified Technique (*n* = 9)	Traditional Technique (*n* = 64)	*p* Value
Patch material used			0.3867
Pericardium patch	5 (55.6)	37 (57.8)	
Modified patch	4 (44.4)	15 (23.4)	
Artificial patch	0 (0.0)	11 (17.2)	
CPB time (min)	124.1 ± 31.8 114.0 (103.0, 171.0)	115.4 ± 69.5 95.0 (81.5, 111.0)	0.0712
Cross-clamp time (min)	89.0 ± 25.1 81.0 (77.0, 98.0)	76.6 ± 33.2 66.0 (55.0, 89.0)	0.0780
Reoperation	0 (0.0)	2 (3.1)	0.5908
**Primary safety outcome**			
Surgery-related complications *	0 (0.0)	14 (21.8)	0.1294
Repeated aortic clamping	0 (0.0)	4 (6.3)	0.4405
Cardiac defibrillation	0 (0.0)	11 (17.2)	0.2090
Delayed chest closure	0 (0.0)	1 (1.6)	0.7057
ECMO needed	0 (0.0)	1 (1.6)	0.7057
Acute myocardial ischemia	0 (0.0)	0 (0.0)	NA
Malignant arrhythmias	0 (0.0)	1 (1.6)	0.7057
**Cardiac echocardiography**			
EF (%)	67.2 ± 5.7 65.0 (63.0, 70.0)	66.4 ± 8.0 65.8 (63.7, 70.0)	0.9060
Supravalvar aortic gradients (mmHg)	17.3 ± 11.2 16.0 (8.4, 18.7)	23.4 ± 17.9 19.7 (10.2, 31.4)	0.4061
Aortic root (mm)	13.8 ± 2.9 14.5 (13.0, 14.5)	13.8 ± 2.7 14.0 (12.0, 15.0)	1.0000
Aortic root z-score	0.6 ± 1.1 0.3 (−0.3, 1.0)	−0.2 ± 1.4 −0.2 (−1.0, 0.8)	0.1514
STJ (mm)	17.5 ± 3.9 18.7 (16.0, 18.7)	18.1 ± 4.0 18.0 (15.0, 19.0)	0.8989
STJ z-score	1.6 ± 1.7 1.1 (0.9, 2.8)	1.2 ± 1.8 1.2 (−0.2, 2.4)	0.6688
Ascending aorta (mm)	16.8 ± 4.2 17.1 (13.0, 17.1)	16.3 ± 4.5 16.0 (13.0, 18.0)	0.4433
Ascending aorta z-score	−0.7 ± 1.3 −0.4 (−1.6, 0.3)	−0.6 ± 1.4 −0.8 (−1.5, 0.4)	0.9933

* Surgery-related complications that included repeated aortic clamping, cardiac defibrillation, delayed chest closure, extracorporeal membrane oxygenation (ECMO) needed, acute myocardial ischemia, malignant arrhythmias. Abbreviation: CPB, extracorporeal circulation; ECMO, extracorporeal membrane pulmonary oxygenation; EF, ejection fraction; STJ, sinotubular junction.

**Table 3 jcm-12-01731-t003:** Follow-up characteristics of the patients with modified and traditional techniques.

Variables	Modified Technique (*n* = 9)	Traditional Technique (*n* = 63)	*p* Value
Follow-up duration (m)	25.0 (12.0, 50.0)	23.0 (3.0, 46.0)	0.9210
EF (%)	67.8 ± 4.2 69.0 (65.0, 70.0)	67.3 ± 4.9 67.2 (65.0, 70.0)	0.6832
Supravalvar aortic gradients (mmHg)	16.5 ± 17.6 7.8 (4.8, 26.0)	22.1 ± 30.3 10.2 (5.0, 23.0)	0.6414
Restenosis	1 (11.1)	9 (14.3)	0.7706
Aortic root (mm)	16.4 ± 4.3 17.0 (15.0, 18.0)	15.1 ± 2.9 15.0 (13.0, 16.1)	0.3158
Aortic root z-score	−0.0 ± 1.0 0.1 (−0.6, 0.7)	−0.4 ± 1.2 −0.5 (−1.3, 0.4)	0.2845
STJ (mm)	21.4 ± 5.9 23.0 (17.0, 26.0)	20.4 ± 4.1 20.1 (17.0, 22.0)	0.4138
STJ z-score	1.1 ± 1.0 1.0 (0.5, 1.6)	1.4 ± 1.8 1.2 (0.3, 2.4)	0.6482
Ascending aorta (mm)	19.1 ± 6.6 19.9 (17.0, 24.0)	18.5 ± 5.2 19.0 (15.0, 21.1)	0.5679
Ascending aorta z-score	−0.4 ± 2.3 0.3 (−0.4, 0.3)	−0.4 ± 2.1 −0.4 (−1.9, 1.1)	0.7789
AVR	0 (0.0)	5 (7.9)	0.5023
**Primary effectiveness outcome**			
Reoperation	0 (0.0)	5 (7.9)	0.5023

Abbreviation: AVR, aortic valve regurgitation; EF, ejection fraction; STJ, sinotubular junction.

## Data Availability

The datasets generated during and/or analyzed during the current study are not publicly available. This dataset will continue to be used in subsequent studies but is available from the corresponding author upon reasonable request.

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
