# Peer review of "Clinical Results of a Modified Doty’s Technique for Supravalvular Aortic Stenosis"

_jcm, 2023, doi:10.3390/jcm12051731_

Round 1
Reviewer 1 Report
- In table 2 which test was used for the analysis of the primary endpoint (Primary safety outcome)? Shouldn't a Fisher's exact test be used? Why did you use the Likelihood Ratio Chi-Square (see attachment)?
- In the method section the authors write "the Chi-squared test or Fisher's exact test was used to compare the categorical data" but it is not clear which test was used, please report for each variables which test was used.
- Please check the p-value calculations for all variables
- P-values should be reported with 4 digits after the decimal point
- Was there missing data? How was this dealt with?
- The primary endpoint definition is somewhat arbitrary, could you provide some references? Was the composite endpoint used as a solution to maintain study feasibility due to low event rates? Are the component endpoints of similar importance to patients?
- The power analysis is completely absent, have you tried to calculate the power of observing the expected value with a sample size of 73 patients?
- Why was no adjusted (multivariable) analysis performed?

Author Response
Dear reviewer 1,
Thanks very much for your thoughtful comments. Changed parts are highlighted in red font in the text. The replies are the following.
Comment 1: In table 2 which test was used for the analysis of the primary endpoint (Primary safety outcome)? Shouldn't a Fisher's exact test be used? Why did you use the Likelihood Ratio Chi-Square (see attachment)?
Response: Thanks for your kind suggestion. We used Likelihood Ratio Chi-Square because the cell frequencies were less than 5. After comparing several methods, we noticed this method has less power. So, we change to Fisher's exact test.
Change: Please see the modified one in Table 2 (Line 166).
Comment 2: In the method section the authors write "the Chi-squared test or Fisher's exact test was used to compare the categorical data" but it is not clear which test was used, please report for each variable which test was used.
Response: We used Fisher's exact test to compared the categorical data.
Change: Please see the modified one in method section (Line 124-125) and Table 1-3 (Line 142, 166, 183).
Comment 3: Please check the p-value calculations for all variables
Response: Thanks for your advice. We have checked p-value calculations for all variables.
Comment 4: P-values should be reported with 4 digits after the decimal point.
Response: Thanks for your advice. All P-values have been corrected.
Change: Please see the modified one in Table 1-3 (Line 142, 166, 183).
Comment 5: Was there missing data? How was this dealt with?
Response: There was no missing data, except that one patient in the traditional technique group lost to follow-up. And we deleted him in the analysis (Table 3).
Comment 6: The primary endpoint definition is somewhat arbitrary, could you provide some references?
Response: Thanks for your advice. We have added explanations for the definitions of the primary endpoint in the Methods section.
Change: Please see the modified one in Line 112-116.
Comment 7: Was the composite endpoint used as a solution to maintain study feasibility due to low event rates? Are the component endpoints of similar importance to patients?
Response: Thank you for your comments. Malignant arrhythmias, acute myocardial ischemia, repeated aortic clamping, cardiac defibrillation, delayed chest closure, and ECMO were all manifestations of hemodynamic instability, low cardiac output, bleeding, arrhythmias, and therapeutic measures after pediatric open heart surgery1, which in this study demonstrated postoperative cardiac insufficiency in children. We used a composite outcome for two reasons:1) we concluded based on previous clinical experience; 2) the importance of the impact of each event on pediatric cardiac surgery is similar; and 3) the low incidence of events Therefore, we combined the events into a composite outcome for analysis.
Comment 8: The power analysis is completely absent, have you tried to calculate the power of observing the expected value with a sample size of 73 patients?
Response: Thanks for your kind advice. We have calculated the sample size of the study. At least 48 patients should be enrolled in each group to ensure 80% power and to detect the difference at the two-sided 5% significance level. Due to the rarity of this disease, after seven years of clinical experience, we collected 64 consecutive supravalvular aortic stenosis patients at two centers, among them, seven were treated with modified Doty's technique by a single operator. Considering the good surgical results, we want to show the exact efficacy of this modified technique for clinicians in this field. Therefore, the aim of this study was to explore the safety and effectiveness of the modified technique.
Comment 9: Why was no adjusted (multivariable) analysis performed?
Response: Thanks for your kind suggestion. We tried to conduct the multivariable analyses. However, due to no event in the modified technique group, the 95% confidence interval of OR was 0.000 to 999.999. Therefore, we did not show this result.
References:
- Denault AY, Deschamps A, Couture P. Intraoperative hemodynamic instability during and after separation from cardiopulmonary bypass. Semin Cardiothorac Vasc Anesth. Sep 2010;14(3):165-82. doi:10.1177/1089253210376673

Reviewer 2 Report
Wang et al. present very interesting data on the surgical treatment of congenital supravalvular aortic stenosis.
The results are of interest to a broad readership and the fact that the introduced operation technique is superior to the traditional operation makes the manuscript even more convincing.
However, I have some concerns about the manuscript which should be solved prior to publication:
- Abstract: Please include very brief information of the change in the operation technique/the technique of modification to the authorship
- Could you give the reader an impression on what basis you decided either for the traditional or the modified technique, e.g. patient characteristics, pecularities of the aortic root or aortic valve?
- In general: Please improve the English level, especially for adjectives/adverbs
- Also check for style errors (letter size, paragraphs...)
Author Response
Dear reviewer 2,
Thanks very much for your thoughtful comments. Changed parts are highlighted in red font in the text. The replies are the following.
Comment 1: Abstract: Please include very brief information of the change in the operation technique/the technique of modification to the authorship.
Response: Thanks for your suggestion. We have added a description of the operative changes in the abstract.
Change: Please see the modified one in line 28-30.
Comment 2: Could you give the reader an impression on what basis you decided either for the traditional or the modified technique, e.g. patient characteristics, pecularities of the aortic root or aortic valve?
Response: Thanks for your advice. We have added a description of the choice of surgical technique to the discussion.
Change: Please see the modified one in line 227-229.
Comment 3: In general: Please improve the English level, especially for adjectives/adverbs.
Response: Thanks for your suggestion. The article has been revised by native English-speaking editors.
Comment 4: Also check for style errors (letter size, paragraphs...)
Response: Thanks for your suggestion. We have checked for full text style errors.

Reviewer 3 Report
This is comparison of new surgical technique and conventional technique.
The number of patients were relatively small, compared to the number of patients who were treated with conventional technique.
But, the result demonstrated that the new technique is safe enough to apply to patients.
Author Response
Dear reviewer 3,
Thanks very much for your thoughtful comments. Changed parts are highlighted in red font in the text. The replies are the following.
Comment 1: This is comparison of new surgical technique and conventional technique.
The number of patients were relatively small, compared to the number of patients who were treated with conventional technique. But the result demonstrated that the new technique is safe enough to apply to patients.
Response: Thank you for your insightful comments. Given that SVAS is a remarkably uncommon form of obstruction in the left ventricular outflow tract, we will persistently follow the population in this study and keep enrolling individuals for the modified procedure.

Round 2
Reviewer 2 Report
The authors have improved the clarity of the manuscript structure. The manuscript has scientific merit so that it seems now suitable for publication.